# Appraisal of Allostatic Load in Wild Boars Under a Controlled Environment

**DOI:** 10.3390/vetsci12070667

**Published:** 2025-07-16

**Authors:** Nadia Piscopo, Anna Balestrieri, Nicola D’Alessio, Pasqualino Silvestre, Giovanna Bifulco, Alessio Cotticelli, Tanja Peric, Alberto Prandi, Danila d’Angelo, Francesco Napolitano, Luigi Esposito

**Affiliations:** 1Dipartimento di Medicina Veterinaria e Produzioni Animali, Università di Napoli Federico II, 80137 Napoli, Italy; giovanna.bifulco@unina.it (G.B.); alessio.cotticelli@unina.it (A.C.); danidange72@gmail.com (D.d.); francesco.napolitano3@unina.it (F.N.); luigi.esposito4@unina.it (L.E.); 2Dipartimento di Sicurezza Alimentare, Istituto Zooprofilattico Sperimentale del Mezzogiorno, 80055 Portici, Italy; anna.balestrieri@izsmportici.it; 3Dipartimento di Sanità Animale, Istituto Zooprofilattico Sperimentale del Mezzogiorno, 80055 Portici, Italy; nicola.dalessio@izsmportici.it; 4Dipartimento di Prevenzione, Igiene e delle Produzioni Zootecniche, ASL Napoli 2 Nord, 80027 Napoli, Italy; pasquale.silvestre@lozoodinapoli.it; 5Dipartimento di Scienze Agroalimentari, Ambientali e Animali, Università degli Studi di Udine, 33100 Napoli, Italy; tanja.peric@uniud.it (T.P.); alberto.prandi@uniud.it (A.P.); 6CEINGE-Biotecnologie Avanzate Franco Salvatore, 80145 Napoli, Italy

**Keywords:** animal welfare, bristles, cortisol, allostatic load, wild boars

## Abstract

Mammals often experience stressful and life-threatening conditions; therefore, they implement neuroendocrine, metabolic or behavioral strategies aimed at survival and, if possible, returning to a physiological state. When the need for adaptation is long-lasting it may eventually lead to poor health conditions. Therefore, assessing allostatic load and resilience abilities is gaining attention in wild species, with the purpose of improving their welfare, even in different contexts, rather than their own ones. Wild boars (*Sus scrofa*) are attracting more negative attention nowadays because of urbanization and forest regrowth, which are causing them to spread across Europe. Thus, they represent a serious threat for farmers, crops and people in general. However, findings about the homeostatic control of stress in these wild species are still lacking. In the present study, we sought to investigate allostatic load of wild boars in a controlled environment through the evaluation of cortisol concentration from bristles collected at different time points. Our data highlighted the importance of adapting proper and effective strategies to monitor long-term stressful events, as well as preserve the physiological conditions of wild boars, and eventually find solutions to conflicts between humans and animal welfare.

## 1. Introduction

In mammals, the concept of animal welfare embraces the existence of strategies adopted with the purpose of coping with stressful events, ensuring a suited lifestyle and maintaining homeostasis across species. As in humans [1], animals can experience either short-term or persistent stressful conditions that bring about a marked increase in the reactivity of neuroendocrine pathways, such as the sympathetic–adrenal–medullary (SAM) and hypothalamus–pituitary–adrenal (HPA) axis [2,3,4]. The SAM and HPA systems cause catecholamines and glucocorticoids to be released, which target multiple organs to help the body adapt to triggering stressors. Resilience to internal and external stressors in animals may depend on the genetics, life experience and behavioral differences between species, which enable them to adapt their physiological responses to allostatic load. However, inadequate stress responses turn into homeostasis dysregulation, which can affect metabolic, immune, inflammatory and cardiovascular outcomes [5]. Different to what is observed in domestic animals, wild animals are more often exposed to stressful stimuli, triggered by the predator–prey relationship, weather conditions, hunting/poaching, food deprivation, toxins, infectious agents or social conflicts, thus representing a concern for species health and conservation [6,7]. Among different biomarkers used to evaluate stress responses in animals, cortisol is considered a reliable measure of allostatic load [8,9,10]. Cortisol concentration has been determined in several biological matrices, including serum, plasma, urine, feces, saliva, milk, feathers, claws and nails [11,12,13,14,15,16,17,18]. To limit the stressors associated with blood sampling, alternative non-invasive matrices, such as hair, have also been used in wild species [19,20,21,22,23,24,25]. Hair samples can be stored at room temperature before being processed and provide useful information about the allostatic load of individuals whilst not being affected by circadian variations of hormone concentrations [26,27,28,29,30]. The interest in the wild boar (*Sus scrofa*) is related to the increase in populations throughout Europe [31,32]. Almost all of the existing literature describes its presence in agroecosystems in relation to the enormous damage they cause to crops [33,34,35], the invasion of urban centers, traffic accidents [35,36,37] and, more recently, the occurrence of African swine fever [38]. Nevertheless, less research has been conducted on the physiological status, resilience and allostatic load of individual animals due to the difficulties in sampling blood matrices (serum, plasma) and handling wild boar. The Campania Region (Southern Italy) has adopted a “Management and control plan for the wild boar species in the designated hunting area” (DGR no. 521 of 23/11/2021) to curb the invasion of wild boar. In accordance with the law, this control can be exercised selectively, minimizing the impact on other species through specific shooting or trapping plans that involve the collaboration of foresters, municipal guards and voluntary hunting guards from associations under the direct control of public veterinarians. Hence, the knowledge gap regarding the allostatic load of wild boars and the opportunity offered by the Regional Wild Boar Containment Plan led to a study of cortisol concentrations in the bristles of wild boars in a controlled environment in the Campania Region (Italy).

## 2. Materials and Methods

### 2.1. Study Area

The present study refers to wild boars captured in the territory of the Campania Region and relocated to the regional forest “Cerreta Cognole” (40°14′44″ N–15°89′31″ E). This facility is suitable and authorized for the reception of live wild boar and organizes veterinary health checks.

### 2.2. Rearing and Observation Area

This study was conducted in the “Cerreta Cognole” forest, a state-owned area of the Campania Region. The environmental and climatic conditions did not differ significantly from the origin area of the animals. The perimeter of the forest was completely fenced (27 km) and the entire area was divided into four separate zones (approximately 200 ha). A specific area of approximately 1 ha was reserved for the animals involved in the experimentation. The surface was equipped with different plant communities, among which the most important is *Quercus cerris*, as well as watering points and feeders. The animals were fed a simple compound feed distributed in a trough and had free access to water. These conditions were maintained constantly throughout the experimental period. On the days when the bristle samples were taken, the wild boars were led into an enclosure of about 40 m^2^ made of chestnut planks and wire mesh by offering them food. Each animal was then led into a corridor that ended in a rectangular cage with a movable panel that was safe for both the operators and the animals (200 × 100 × 100 cm). At the end of the sample collection operations, the animals could freely access the area, ensuring the conditions of a controlled environment.

### 2.3. Animals and Sampling

Each animal was identified upon arrival in the housing and health monitoring area by a microchip implanted under the skin of the left shoulder. The individual information of each sampled animal was recorded and included date, time, type of capture, gender, pregnancy status, estimated age, weight, clinical examination and any anomalies.

The capture campaigns allowed the formation of a homogeneous group of non-pregnant wild female boars, allocated into three age groups, i.e., young (G1, <10 months), sub-adults (G2, 10–12 months) and adults (G3, >12 months), according to the criteria described by Piscopo et al. [19] and Güldenpfennig et al. [32].

A total of 108 bristle samples were collected from 18 different wild boars during the six-month observation period (January–June) in which the animals were kept under identical conditions. On the day the animals arrived in the housing and health observation area (20 days before the start of the test), the first cut of the bristles was made at the left shoulder. This procedure was considered time zero and was sufficient to allow regrowth of the bristles from the subepidermal layer. Later, the bristles were collected by trained veterinarians, following standard containment procedures, approximately once a month from the same area using the shave–reshave method. The clipped bristles were placed in a paper bag labelled with the animal’s microchip and the date of collection and were stored in a dark at room temperature.

### 2.4. Laboratory Analysis

Bristle strands were washed and extracted as described by Bergamin et al. [39]. Washing with isopropanol is essential to minimize the risk of extracting steroids from the surface of the bristle, which have been deposited by sweat and sebum. The concentrations of cortisol were measured using an in-house enzyme-linked immunosorbent assay (ELISA), as described already for human hair by Falco et al. [40].

### 2.5. Statistical Analysis

The statistical analyses were performed in R version 4.4.3 for Windows 11. The normal distribution and homogeneity of variances of the data were verified using the Shapiro–Wilk and Levene test, respectively. First, an analysis of variance (ANOVA) for repeated measures was conducted to evaluate the effects of age as a fixed factor and sampling month as repeated measure. Cortisol concentrations were the dependent variable. The ANOVA model was built using the aov() function. Then, individual months were compared between and within age groups using post-hoc pairwise comparisons. Tukey’s honest significant difference (HSD) and Bonferroni adjustments were applied to control multiple comparisons, implemented using the emmeans package.

Data were visualized using the ggplot2 package in R (R version 4.4.3 for Windows 11). The significance level was set at *p* < 0.05.

### 2.6. Ethical Note

The authors operated with the greatest respect for the behavior of the *Sus scrofa* species and the approach to this study was ethically correct. The approach for managing animals captured from the wild was to respect the five freedoms linked to animal welfare as much as possible, guaranteeing them:(1)freedom from hunger, thirst and poor nutrition (water ad libitum and daily distribution of a complete diet);(2)freedom from environmental disturbances (the animals were housed in a fenced wooded area);(3)freedom from disease and injury (monthly veterinary check-up);(4)freedom to freely express species-specific behavioral characteristics (adequate spaces in which to develop the correct intraspecific competitions necessary for the formation of the family group);(5)freedom from fear and stress (less anthropogenic disturbance during the entire observation period; evaluation of HPA axis activity through cortisol concentrations).

The present study involved wild animals kept in captivity, captured in accordance with the law using specific traps for wild boars (cages with platform deception and guillotine closure; frames made of removable galvanized sheet metal profiles and galvanized netting with 30 × 30 mm mesh, which does not cause physical harm to the animals; dimensions 100 × 100 × 200 cm). It should be noted that these capture cages cannot be sold to private individuals but only to public entities for activities foreseen pursuant to current legislation (art. 4; art. 10, paragraph 7; art. 19, paragraph 2; art. 19-bis of Law no. 157/1992, as well as art. 11, paragraph 4 and art. 19, paragraph 5 of Law 394/1991).

## 3. Results and Discussion

Here, we evaluated the 30-day cortisol (CORT) production integrated into the bristles of the wild boars enrolled in this study over six months (January–June 2022). The ANOVA indicated an overall effect of age (F-value= 14.83, *p* < 0.0001). Particularly, multiple comparisons showed significantly higher concentrations of CORT in young subjects (G1) compared to sub-adults (G2) and adults (G3). As a matter of fact, the comparison between the CORT values of juveniles (G1) vs. sub-adults (G2) was significantly different (3.90 ± 0.21 vs 2.57 ± 0.25 pg/mg, *p* < 0.0001; as well as the comparison between juveniles (G1) vs. adults (G3): 3.90 ± 0.21 vs. 2.83 ± 0.25 pg/mg, *p* = 0.0002; Table 1).

Bristle CORT concentrations in the wild boars were further investigated at sequential sampling time points during their stay in the controlled environment (Figure 1). In particular, multiple comparisons tests showed higher CORT concentrations in young subjects (G1) at the first sampling time point (January) compared to both adult and sub-adult groups (*p* < 0.05).

In the present preliminary work, we assessed CORT concentrations in the bristles of wild boars, which were temporary kept in a controlled environment to monitor and manage their health status. Our findings describe the physiological values of this stress-related hormone in wild boars, ranging from 1.49 to 7.68 pg/mg, which are consistent with previous works conducted on the same species [41,42,43]. The animals enrolled in this study were caught at the end of the three-month hunting season and were subjected to significant allostatic load during the following six months. In our study, the animals were successfully grouped into three homogeneous groups characterized by different ages, which is usually challenging when dealing with wildlife species. Young (G1) boars showed overall higher total CORT concentrations compared to both sub-adult (G2) and adult (G3) subjects, which was in line with previous data on domestic pigs [44,45]. On the other hand, studies conducted on wild boars [41,42] did not investigate young animals, while unfortunately, Tajchman et al. (2024) [43] combined cortisol data from wild boars (young males aged 1–3 years) with data from another wild species. The monthly evaluation of allostatic load throughout the experimental period highlighted higher CORT concentrations in young animals (G1) in January and February, which then declined by May. It seems likely that the highest CORT concentration described in January (4.88 ± 0.71 pg/mg) is related to the physiologically higher perinatal bristle CORT concentrations, whereas the levels measured in re-growth bristles in February (5.14 ± 0.54 pg/mg) reflect adaptation processes to the new environment. Most interestingly, the effect of the time was not significant in both sub-adult and adult groups, indicating a smoother adaptability to new conditions and management. This latter result allows us to hypothesize that, in sufficiently safe and constant environmental conditions over time, young animals express good potential for adaptation too. As a matter of fact, several data are available on the stress response of wild species to captivity, where they are exposed to environmental and weather influences, diseases, habitat fragmentation and loss and urban stressors [46,47]. In our study, the boars were kept in a controlled environment with free, controlled access to food, highlighting the importance of having suitable areas to better understand their physiological needs and improve management strategies for their conspecifics. Although further studies involving a wider cohort of animals and possibly during a broader timeframe are needed, our preliminary data allowed us to evaluate the adaptation of wild boars to different and new environmental conditions. Since cortisol concentrations in the bristle matrix integrate retrospective information regarding the allostatic load of the animals [18,44,48], our preliminary results provide reference values that may contribute to the growing knowledge about wild boars and help fill gaps in evaluating animal welfare, especially concerning the conservation and management of wild species.

## Figures and Tables

**Figure 1 vetsci-12-00667-f001:**
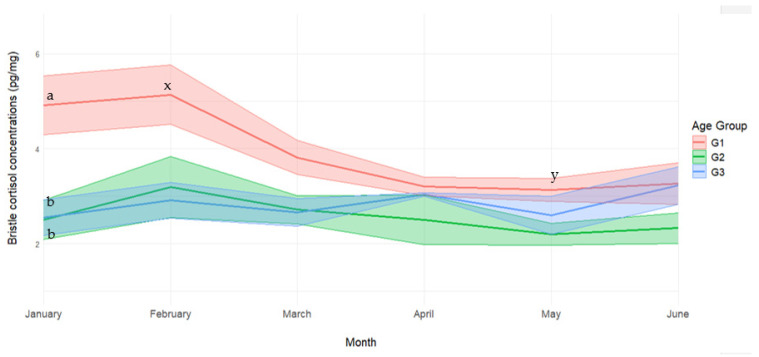
Bristle cortisol concentrations (pg/mg) grouped by age throughout the trial. ^a,b^ represents differences between groups at *p* < 0.05. ^x,y^ represents differences within groups at *p* < 0.05.

**Table 1 vetsci-12-00667-t001:** Cortisol concentration (pg/mg) in the bristles of the wild boars in the three different groups (G1, G2, G3), collected monthly over a period of 180 days.

Age	Mean	SE
G1	3.90 ^B^	0.21
G2	2.57 ^A^	0.25
G3	2.83 ^A^	0.25

All values are expressed as estimated marginal means ± standard error (SE). ^A,B^ represents a difference at *p* < 0.01.

## Data Availability

Dataset available on request from the authors.

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
