# Peer review of "Appraisal of Allostatic Load in Wild Boars Under a Controlled Environment"

_vetsci, 2025, doi:10.3390/vetsci12070667_

Round 1
Reviewer 1 Report
Comments and Suggestions for Authors
This manuscript looks at the cortisol levels of wild boars at 3 different age groups held in captivity over 6 months.
From the introduction, I'm gathering that the country captures these wild pigs for health surveillance and conservation questions regardless if there was a research study on them? So the research was just seeing if captivity for 6 months is stressful to the pigs.
It could be of interest to help answer this question to compare it to cortisol concentrations of wild boars in the wild (either capture and released, or if they do shoot one take a sample immediately). This may help with knowing more about an increase from Jan to Feb to know if that's a seasonal increase in cortisol or they were more stressed the first month of captivity while they adapted to the new environment.
There is a note on ln 199 there was significant allostatic load during the study period, though it may be out of the control of the researchers, the information on the type of procedures to cause 'allostatic load' and how often, when etc in the 6month period may be helpful to understand the whole 6 month timeline, since though not significant, the cortisol concentrations do go up and down in G2 and G3.
Ln 203 - when comparing data to domestic pigs is that also hair cortisol samples or other specimens?
Author Response
Comment 1. This manuscript looks at the cortisol levels of wild boars at 3 different age groups held in captivity over 6 months.
Response 1. The authors confirm that the experimental design is the one reported by referee 1.
Comment 2. From the introduction, I'm gathering that the country captures these wild pigs for health surveillance and conservation questions regardless if there was a research study on them? So the research was just seeing if captivity for 6 months is stressful to the pigs.
Response 2. Referee 1 correctly understood that the wild boars originate from a population control program practiced by the Campania Region, independent of the research study. Thanks to this practice, the authors were able to study a group of animals for six consecutive months, under the same conditions, assessing their stress related to captivity. This situation cannot be replicated in natural conditions.
Comment 3. It could be of interest to help answer this question to compare it to cortisol concentrations of wild boars in the wild (either capture and released, or if they do shoot one take a sample immediately). This may help with knowing more about an increase from Jan to Feb to know if that's a seasonal increase in cortisol or they were more stressed the first month of captivity while they adapted to the new environment.
Response 3. Authors thank referee 1 for the interesting comment and note that wild boar hunting is not permitted during the January-February period, making it difficult to obtain comparison samples. In any case, hunting activities, typical of wild boar, are extremely stressful, and the values obtained from bristle analysis would not reflect acute stress. In previous work authors have investigated this topic, but these studies used blood samples, which are not comparable to the current study.
Comment 4. There is a note on ln 199 there was significant allostatic load during the study period, though it may be out of the control of the researchers, the information on the type of procedures to cause 'allostatic load' and how often, when etc in the 6month period may be helpful to understand the whole 6 month timeline, since though not significant, the cortisol concentrations do go up and down in G2 and G3.
Response 4. Regarding referee 1 for the sentence on line 199, the authors report that: Group G1 consists of animals younger than 10 months of age that can be considered prepubescent. At the end of six months, these animals have reached puberty. Group G2 consists of animals between 10 and 12 months of age, meaning they have already reached puberty in the first months of the experiment. Group G3 consists of animals that are already pubescent at the first month of experimental period. This consideration suggests that G1 animals are not affected by reproductive status. G2 animals differ from G3 in that reproductive status begins and develops during the experimental period, while G3 expresses reproductive status throughout the experimental period. This could explain the different, albeit not statistically significant, responses to allostatic load.
Comment 5. Ln 203 - when comparing data to domestic pigs is that also hair cortisol samples or other specimens?
Response 5. Yes, both papers with references 44 and 45, refers to hair cortisol levels in piglets.

Reviewer 2 Report
Comments and Suggestions for Authors
Comments:
The brief report written by Piscopo et al. investigated allostatic load in wild boars under a controlled environment in the Campania region of Southern Italy. The authors evaluated cortisol concentrations in the bristles of young, sub-adult, and adult female wild boars over six months. The study aimed to assess how well wild boars adapt to a controlled environment and to contribute to the understanding of their physiological stress responses. The study provides valuable preliminary data on allostatic load in wild boars, highlighting the importance of age and adaptation to new environments. However, the small sample size (only 18 single gender wild boars) and short observation period (6 months) indicate that further study with larger, more diverse populations and longer observation periods is needed to validate and expand upon these findings.
Specific comments:
- Relying solely on glucocorticoid (cortisol) concentrations in bristles may not fully capture the complete stress picture in wild boars. Heat shock proteins (HSPs) are a family of proteins that are upregulated in response to various stressors, including heat, oxidative stress, inflammation, and exposure to toxins. They act as molecular chaperones, helping to protect and repair damaged proteins, thus maintaining cellular homeostasis. Suggest detecting the expression levels of HSPs in wild boars without invasive matrices. Combining cortisol measurements with HSPs expression data can help to distinguish between short-term and chronic stress, as well as to assess the animal's capacity to cope with stressors.
- The lack of a free-ranging control group makes it difficult to definitively attribute the observed cortisol levels to the controlled environment.
- Replace the figure with a higher resolution one.
Author Response
Comment 1. Relying solely on glucocorticoid (cortisol) concentrations in bristles may not fully capture the complete stress picture in wild boars. Heat shock proteins (HSPs) are a family of proteins that are upregulated in response to various stressors, including heat, oxidative stress, inflammation, and exposure to toxins. They act as molecular chaperones, helping to protect and repair damaged proteins, thus maintaining cellular homeostasis. Suggest detecting the expression levels of HSPs in wild boars without invasive matrices. Combining cortisol measurements with HSPs expression data can help to distinguish between short-term and chronic stress, as well as to assess the animal's capacity to cope with stressors.
Response 1. We thank the reviewer for her/his thoughtful raised comment. Several findings documented a correlation between thermal stress and altered expression levels of such chaperone-like proteins, thus making them ideal candidates to delineate the impact of environmental thermal stress, especially on dairy animals. accordingly, in a very recent paper the authors showed a strong correlation of salivary HPS 70 with hematobiochemical, endocrinological, physiological, behavioural, nutritional, and milk production responses in thermally stressed dairy cows (Rajamanickam K et al., 2025). So, although we recognize the importance of evaluating HSP as a reliable thermal stress-related biomarker, unfortunately we do not have samples from the animals to address this issue. However, we have to say that, as mentioned in the Methods section of the manuscript, the environmental and climatic conditions of the observation area of Cerreta Cognole did not differ significantly from their origin area of the analyzed wild boars, and we were unable to assess the impact on HSP expression levels.
Comment 2. The lack of a free-ranging control group makes it difficult to definitively attribute the observed cortisol levels to the controlled environment.
Response 2. We thank the reviewer for her/his consideration, on which we agree. Given the objective difficulties to collect sample from free-ranging animals, in the present study we could just compared CORT levels at different age groups and, despite lacking proper experimental control groups, the present study tried to document, on one hand, the higher CORT levels in G1 (young) animals (maybe expected), at the very moment when they were introduced in the fenced area (between January and February) and, on the other, their ability to adapt to the novel (most likely not as stressful as their physiological context) environment.
Comment 3. Replace the figure with a higher resolution one.
Response 3. In the revised version of the manuscript we added a figure with a higher resolution.

Round 2
Reviewer 2 Report
Comments and Suggestions for Authors
No further comment.